# Spectral technique with convergence analysis for solving one and two-dimensional mixed Volterra-Fredholm integral equation

A. Z. Amin[1], A. K. Amin[2,3], M. A. Abdelkawy[3,4]*, A. A. Alluhaybi[2], I. Hashim[1]

1 Department of Mathematical Sciences, Faculty of Science & Technology, Universiti Kebangsaan, Malaysia, Malaysia, 2 Department of Basic Sciences, Adham University College, Umm AL-Qura University, Makkah, Saudi Arabia, 3 Department of Mathematics, Faculty of Science, Beni-Suef University, Beni-Suef, Egypt, 4 Department of Mathematics and Statistics, College of Science, Al-Imam Mohammad Ibn Saud Islamic University (IMSIU), Riyadh, Saudi Arabia

* maohamed@imamu.edu.sa

**Data Availability Statement:** All relevant data are within the paper.

**Funding:** The author(s) received no specific funding for this work.

## Abstract

A numerical approach based on shifted Jacobi-Gauss collocation method for solving mixed Volterra-Fredholm integral equations is introduced. The novel technique with shifted Jacobi-Gauss nodes is applied to reduce the mixed Volterra-Fredholm integral equations to a system of algebraic equations that has an easy solved. The present algorithm is extended to solve the one and two-dimensional mixed Volterra-Fredholm integral equations. Convergence analysis for the present method is discussed and confirmed the exponential convergence of the spectral algorithm. Various numerical examples are approached to demonstrate the powerful and accuracy of the technique.

## 1 Introduction

Integral equations (IEs) are regarded as one of the most important types of equations. Mixed Volterra-Fredholm integral equations (MV-F-IEs) [1–4] arise in various physical and biological problems and play a substantial role in describing real-life phenomena in other areas of science. Some IEs also appear in other applications, including the theory of parabolic initial boundary value problems, the spatio-temporal growth of an epidemic, Fourier problems, population dynamics, population genetics, mechanics, molecular conduction, and biological problems. The two-dimensional IEs have many applications including telegraph model [5], plasma physics [6] and electrical engineering [7]. Therefore, the study of these types of IEs and the creation and implementation of analytical and numerical methods for solving them are hot areas of research. Analytical solutions for IEs either do not exist or are difficult to achieve. In light of the preceding discussion, numerical techniques have been proposed and developed for approximating solutions of IEs.

Various numerical techniques have been proposed for solving the one-dimensional IEs, for example [8–10]. The authors in [11] introduced a numerical scheme for the solution of MV-F-IEs by means of the moving least square method together with Chebyshev polynomials. The authors in [12] applied Legender collocation technique to obtain an accurate numerical

**Competing interests:** The authors have declared that no competing interests exist.

solution of MV-F-IEs. Whilst in [13], the multiquadric radial basis functions have been investigated to numerically solve the MV-F-IEs. Recently, in [14], the authors proposed and developed the triangular functions method to study two-dimensional MV-F-IEs. In the same line, the authors in [15, 16] used Taylor and Lagrange collocation technique to acquire the numerical solutions of MV-F-IEs. Moreover, other numerical solutions of MVF-IEs were studied in [17–19].

Spectral methods [20–23] have been widely utilized in several areas in the last four decades. In the early times, spectral techniques based on Fourier expansion have applied in a few scopes, such as a simple geometric area and periodic boundary conditions. Newly, they have sophisticated theoretically and utilised as powerful techniques to dissolve different kind of problems including linear and nonlinear differential equations, integral, integro-differential equations, fractional differential equations, optimal control, and variational problems. Based on the thoroughness and exponential averages of convergence, the spectral techniques have a premium character when compared with other numerical techniques. The expression of the problem solution as a finite series of several functions is the fundamental step of all kinds of spectral techniques. Then, the coefficients will be chosen such the absolute error is minimized as well as possible. Whilst, the numerical solution will be implemented almost to satisfy differential equations in spectral collocation technique [24–29]. On the other hand, the residuals may be permitted to be zero at selection points. The collocation approach has been successfully applied in a wide range of scientific and engineering areas due to its obvious advantages. Because their global nature fits well with the nonlocal definition of fractional operators, spectral collocation methods are promising candidates for solving fractional differential equations.

In this paper, numerical solutions of the one and two-dimensional MV-F-IEs are obtained by means of the shifted Jacobi-Gauss collocation (SJ-G-C) scheme. The solution of a such equation is approximated as a finite expansion of shifted Jacobi polynomials (SJP) for independent variables, and then we evaluate the residual of the MV-F-IEs at the shifted Jacobi-Gauss (SJ-G) quadrature points. The novel method together with the SJ-G nodes is utilized to reduce the MV-F-IEs to a system of algebraic equations that can be easily solved. We also analyze the convergence of the present method. Numerical simulations for some MV-F-IEs are presented to demonstrate the accuracy of the method.

This paper is organized as follows: In Section 2, we present a few preliminary remarks and some information about SJP. Section 3 is divided into two subsections: one for solving one-dimensional linear MV-F-IEs using a new collocation method, and the other for numerically solving one-dimensional non-linear MV-F-IEs. In Section 4, the novel algorithm is expanded to solve two-dimensional MV-F-IEs. The error analysis of the proposed algorithm is discussed in Section 5. Numerical simulations are presented in Section 6 to ensure the effectiveness and accuracy of the technique. Section 7 outlines some observations and conclusions.

## 2 Properties of shifted Jacobi polynomials

Some properties of SJP are introduced in this section. Jacobi polynomials are given by [30]

$$P_{k+1}^{(\rho,\gamma)}(y) = (a_k^{(\rho,\gamma)}y - b_k^{(\rho,\gamma)})P_k^{(\rho,\gamma)}(y) - c_k^{(\rho,\gamma)}P_{k-1}^{(\rho,\gamma)}(y), k \geq 1,$$

$$P_0^{(\rho,\gamma)}(y) = 1, P_1^{(\rho,\gamma)}(y) = \frac{1}{2}(\rho + \gamma + 2)y + \frac{1}{2}(\rho - \gamma),$$

$$P_k^{(\rho,\gamma)}(-y) = (-1)^k P_k^{(\rho,\gamma)}(y), P_k^{(\rho,\gamma)}(-1) = \frac{(-1)^k \Gamma(k + \gamma + 1)}{k!\Gamma(\gamma + 1)}, \tag{2.1}$$

wherever $\rho, \gamma > -1$, $y \in [-1, 1]$ and

$$
a_k^{(\rho,\gamma)} = \frac{(2k + \rho + \gamma + 1)(2k + \rho + \gamma + 2)}{2(k+1)(k + \rho + \gamma + 1)},
$$

$$
b_k^{(\rho,\gamma)} = \frac{(\gamma^2 - \rho^2)(2k + \rho + \gamma + 1)}{2(k+1)(k + \rho + \gamma + 1)(2k + \rho + \gamma)},
$$

$$
c_k^{(\rho,\gamma)} = \frac{(k + \rho)(k + \gamma)(2k + \rho + \gamma + 2)}{(k+1)(k + \rho + \gamma + 1)(2k + \rho + \gamma)}.
$$

Furthermore, the $n$th derivative ($n$ is an intger) of $P_j^{(\rho,\gamma)}(y)$, may be acquired from [30]

$$
D^n P_j^{(\rho,\gamma)}(y) = \frac{\Gamma(j + \rho + \gamma + q + 1)}{2^n \Gamma(j + \rho + \gamma + 1)} P_{j-n}^{(\rho+n,\gamma+n)}(y). \tag{2.2}
$$

For the SJP $P_{L,k}^{(\rho,\gamma)}(y) = P_k^{(\rho,\gamma)}(\frac{2y}{L} - 1)$, $L > 0$, the straightforward analytic form is written as

$$
\begin{aligned}
P_{L,k}^{(\rho,\gamma)}(y) &= \sum_{j=0}^{k} (-1)^{k-j} \frac{\Gamma(k + \gamma + 1)\Gamma(j + k + \rho + \gamma + 1)}{\Gamma(j + \gamma + 1)\Gamma(k + \rho + \gamma + 1)(k-j)!j!L^j} y^j \\
&= \sum_{j=0}^{k} \frac{\Gamma(k + \rho + 1)\Gamma(k + j + \rho + \gamma + 1)}{j!(k-j)!\Gamma(j + \rho + 1)\Gamma(k + \rho + \gamma + 1)L^j} (y - L)^j.
\end{aligned} \tag{2.3}
$$

Therefore, we can derive the next properties for any integer $n$

$$
\begin{aligned}
P_{L,k}^{(\rho,\gamma)}(0) &= (-1)^k \frac{\Gamma(k + \gamma + 1)}{\Gamma(\gamma + 1)\ k!}, \\
P_{L,k}^{(\rho,\gamma)}(L) &= \frac{\Gamma(k + \rho + 1)}{\Gamma(\rho + 1)\ k!},
\end{aligned} \tag{2.4}
$$

$$
D^n P_{L,k}^{(\rho,\gamma)}(0) = \frac{(-1)^{k-n}\Gamma(k + \gamma + 1)(k + \rho + \gamma + 1)_n}{L^n \Gamma(k - n + 1)\Gamma(n + \gamma + 1)}, \tag{2.5}
$$

$$
D^n P_{L,k}^{(\rho,\gamma)}(L) = \frac{\Gamma(k + \rho + 1)(k + \rho + \gamma + 1)_n}{L^n \Gamma(k - n + 1)\Gamma(n + \rho + 1)}, \tag{2.6}
$$

$$
D^n P_{L,k}^{(\rho,\gamma)}(y) = \frac{\Gamma(n + k + \rho + \gamma + 1)}{L^n \Gamma(k + \rho + \gamma + 1)} P_{L,k-n}^{(\rho+n,\gamma+n)}(y). \tag{2.7}
$$

Assuming that $w_L^{(\rho,\gamma)}(y) = (L - y)^\rho y^\gamma$, we can determine the norm and inner product for the weighted space $L_{w_L^{(\rho,\gamma)}}^2[0, L]$ as [30]

$$
(u, v)_{w_L^{(\rho,\gamma)}} = \int_0^L \Lambda(y)\, v(y)\, w_L^{(\rho,\gamma)}(y)\, dy,\ \|v\|_{w_L^{(\rho,\gamma)}} = (v, v)_{w_L^{(\rho,\gamma)}}^{\frac{1}{2}}. \tag{2.8}
$$

The collection of SJP forms a complete $L^2_{w_L^{(\rho,\gamma)}}[0, L]$-orthogonal system. Furthermore, and as a result of (2.8), we have

$$\|P_{L,k}^{(\rho,\gamma)}\|^2_{w_L^{(\rho,\gamma)}} = \left(\frac{L}{2}\right)^{\rho+\gamma+1} h_k^{(\rho,\gamma)} = h_{L,k}^{(\rho,\gamma)}. \tag{2.9}$$

We utilized $y_{N,j}^{(\rho,\gamma)}$, and, $\varpi_{N,j}^{(\rho,\gamma)}$, $0 \leqslant j \leqslant N$, as the nodes and Christoffel numbers on the interval $[-1, 1]$. The corresponding nodes and corresponding Christoffel numbers of the shifted Jacobi on the interval $[0, L]$ can be given by

$$y_{L,N,j}^{(\rho,\gamma)} = \frac{L}{2}(y_{N,j}^{(\rho,\gamma)} + 1),$$

$$\varpi_{L,N,j}^{(\rho,\gamma)} = \left(\frac{L}{2}\right)^{\rho+\gamma+1} \varpi_{N,j}^{(\rho,\gamma)}, \ \ 0 \leqslant j \leqslant N.$$

For any $\phi \in S_{2N+1}[0, L]$ and utilizing quadrature property, we get

$$
\begin{aligned}
\int_0^L (L-y)^\rho y^\gamma \phi(y) dy &= \left(\frac{L}{2}\right)^{\rho+\gamma+1} \int_{-1}^1 (1-y)^\rho (1+y)^\gamma \phi\left(\frac{L}{2}(y+1)\right) dy \\
&= \left(\frac{L}{2}\right)^{\rho+\gamma+1} \sum_{j=0}^N \varpi_{N,j}^{(\rho,\gamma)} \phi\left(\frac{L}{2}(y_{N,j}^{(\rho,\gamma)} + 1)\right) \\
&= \sum_{j=0}^N \varpi_{L,N,j}^{(\rho,\gamma)} \phi\left(y_{L,N,j}^{(\rho,\gamma)}\right).
\end{aligned}
\tag{2.10}
$$

## 3 One-dimensional of mixed Volterra-Fredholm integral equations

Based on the SJ-G-C technique, two algorithms have derived for the numerical treatment of MV-F-IEs. The main objective of the novel technique is to create the MV-F-IEs, to generate a system of algebraic equations of the unknown coefficients and the system can be solved by Newton's iterative technique.

### 3.1 Linear of mixed Volterra-Fredholm integral equations

In this subsection, we utilize the spectral collocation method for solving one dimensional linear MV-F-IEs in the next form [16],

$$\Lambda(\varrho) = g(\varrho) + \int_0^{h(\varrho)} k_1(\varrho, \sigma)\Lambda(\sigma)d\sigma + \int_0^L k_2(\varrho, \sigma)\Lambda(\sigma)d\sigma, 0 \leq \varrho, \sigma \leq L, 0 \leq h(\varrho) < \infty, \tag{3.1}$$

wherever $g(\varrho)$, $k_1(\varrho, \sigma)$ and $k_2(\varrho, \sigma)$ are given functions. Whilst $\Lambda(\varrho)$ is unknown function.

Now, we are concerned in utilizing the SJ-G-C technique to convert, the preceding MV-F-IEs into a system of algebraic equations. To this end, we approximate the unknown function utilizing SJ-G-C technique at $\varrho_{L,N,j}^{(\rho,\gamma)}$ nodes.

In next form, we outline major steps of the SJ-G-C way to solve one dimensional linear MV-F-IEs. We select the approximate solution to be of the form [30]

$$\Lambda_N(\varrho) = \sum_{j=0}^{N} a_j P_{L,j}^{(\rho,\gamma)}(\varrho). \tag{3.2}$$

As a consequence the approximation (3.2), one can write Eq (3.1) like

$$\sum_{j=0}^{N} a_j P_{L,j}^{(\rho,\gamma)}(\varrho) = g(\varrho) + \int_0^{h(\varrho)} (k_1(\varrho,\sigma)\sum_{j=0}^{N} a_j P_{L,j}^{(\rho,\gamma)}(\sigma))d\sigma + \int_0^{L} (k_2(\varrho,\sigma)\sum_{j=0}^{N} a_j P_{L,j}^{(\rho,\gamma)}(\sigma))d\sigma. \tag{3.3}$$

In this way, the residual of (3.3) is set to zero at $N + 1$ of SJ-G points. Therefore, adopting (3.2)-(3.3), enable one to record (3.1) in the next form:

$$\sum_{j=0}^{N} a_j P_{L,j}^{(\rho,\gamma)}(\varrho_{L,N,i}^{(\rho,\gamma)}) = \int_0^{h(\varrho_{L,N,i}^{(\rho,\gamma)})} (k_1(\varrho_{L,N,i}^{(\rho,\gamma)},s)\sum_{j=0}^{N} a_j P_{L,j}^{(\rho,\gamma)}(\sigma))d\sigma$$

$$+ \int_0^{L} (k_2(\varrho_{L,N,i}^{(\rho,\gamma)},s)\sum_{j=0}^{N} a_j P_{L,j}^{(\rho,\gamma)}(\sigma))d\sigma + g(\varrho_{L,N,i}^{(\rho,\gamma)}), \quad i = 0,\ldots,N, \tag{3.4}$$

then the preceding equation can be rewite, to get

$$\sum_{j=0}^{N} a_j \left( P_{L,j}^{(\rho,\gamma)}(\varrho_{L,N,i}^{(\rho,\gamma)}) - \int_0^{h(\varrho_{L,N,i}^{(\rho,\gamma)})} k_1(\varrho_{L,N,i}^{(\rho,\gamma)},s)P_{L,j}^{(\rho,\gamma)}(\sigma)d\sigma - \int_0^{L} k_2(\varrho_{L,N,i}^{(\rho,\gamma)},s)P_{L,j}^{(\rho,\gamma)}(\sigma)d\sigma \right)$$

$$= g(\varrho_{L,N,i}^{(\rho,\gamma)}), \quad i = 0,\ldots,N. \tag{3.5}$$

The preceding system of algebraic equations that has an easy solved. After the coefficients $a_j$ are specified, it can be easily computed the approximate solution $\Lambda_N(\varrho)$ at any value of $(\varrho)$ in the given domain from the next equation

$$\Lambda_N(\varrho) = \sum_{j=0}^{N} a_j P_{L,j}^{(\rho,\gamma)}(\varrho). \tag{3.6}$$

## 3.2 Non-linear mixed Volterra-Fredholm integral equations

In this subsection, we expand the preceding technique for the numerical treatment of the next nonlinear MV-F-IEs:

$$\Lambda(\varrho) = g(\varrho) + \int_0^{h(\varrho)} k_1(\varrho,\sigma,\Lambda(\sigma))d\sigma + \int_0^{L} k_2(\varrho,\sigma,\Lambda(\sigma))d\sigma, 0 \le \varrho,\sigma \le L, 0 \le h(\varrho) < \infty, \tag{3.7}$$

wherever $g(\varrho)$, $k_2(\varrho,\sigma,\Lambda(\sigma))$ and $k_2(\varrho,\sigma,\Lambda(\sigma))$ are given functions, whilst $\Lambda(\varrho)$ is an unknown function.

Alike steps to that provided in the preceding subsection, permit one to reproduce the problem in the form,

$$\sum_{j=0}^{N} a_j P_{L,j}^{(\rho,\gamma)}(\varrho) = g(\varrho) + \int_{0}^{h(\varrho)} k_1(\varrho, \sigma, \sum_{j=0}^{N} a_j P_{L,j}^{(\rho,\gamma)}(\sigma)) d\sigma + \int_{0}^{L} k_2(\varrho, \sigma, \sum_{j=0}^{N} a_j P_{L,j}^{(\rho,\gamma)}(\sigma)) d\sigma. \quad (3.8)$$

Based on the data inserted in this subsection and the recent one, the residual of (3.7) is set to zero at $N + 1$ of SJ-G points, which given the subsequent form:

$$\sum_{j=0}^{N} a_j P_{L,j}^{(\rho,\gamma)}(\varrho_{L,N,i}^{(\rho,\gamma)}) = \int_{0}^{h(\varrho_{L,N,i}^{(\rho,\gamma)})} k_1(\varrho_{L,N,i}^{(\rho,\gamma)}, \sigma, \sum_{j=0}^{N} a_j P_{L,j}^{(\rho,\gamma)}(\sigma)) d\sigma$$

$$+ \int_{0}^{L} k_2(\varrho_{L,N,i}^{(\rho,\gamma)}, \sigma, \sum_{j=0}^{N} a_j P_{L,j}^{(\rho,\gamma)}(\sigma)) d\sigma + g(\varrho_{L,N,i}^{(\rho,\gamma)}), \quad i = 0, \ldots, N. \quad (3.9)$$

The former nonlinear algebraic system can be solved utilising Newton's iterative method.

$$\Lambda_N(\varrho) = \sum_{j=0}^{N} a_j P_{L,j}^{(\rho,\gamma)}(\varrho). \quad (3.10)$$

## 4 Two-dimensional mixed Volterra-Fredholm integral equations

In this section, the preceding numerical algorithms have extended for numerically solving the linear and non-linear two-dimensional MV-F-IEs. The collocation points are chosen at the SJ-G interpolation nodes. The essence of the introduced technique is to discretize the two-dimensional MV-F-IEs to create a system of algebraic equations of the unknown coefficients.

### 4.1 Linear mixed Volterra-Fredholm integral equations

In subsection, we enlarge the analysis in 3.1 for solving the next two-dimensional linear MV-F-IEs

$$\Lambda(\varrho, \sigma) = \int_{0}^{t} \int_{0}^{L} K(\varrho, \sigma, y, z) \Lambda(y, z) dy dz, + g(\varrho, \sigma) \qquad 0 \leq \varrho, \sigma \leq L, \quad (4.1)$$

wherever $g(\varrho, \sigma)$ and $K(\varrho, \sigma, y, z)$ are a given functions, whilst $\Lambda(\varrho, \sigma)$ is an unknown function. Hence, the SJ-G-C technique will be implemented to convert the preceding two-dimensional MV-F-IEs into the system of algebraic equations.

Let us extend the dependent variable in the model,

$$\Lambda_{N,M}(\varrho, \sigma) = \sum_{i=0}^{M} \sum_{j=0}^{N} a_{ij} P_{L,j}^{(\rho_1,\gamma_1)}(\varrho) P_{\tau,i}^{(\rho_2,\gamma_2)}(\sigma). \quad (4.2)$$

Therefore, adapting (4.2), enable one to write (4.1) as

$$\sum_{i=0}^{M}\sum_{j=0}^{N}a_{ij}P_{L,j}^{(\rho_1,\gamma_1)}(\varrho)P_{\tau,i}^{(\rho_2,\gamma_2)}(\sigma) = \int_{0}^{t}\int_{0}^{L}K(\varrho,\sigma,y,z)\sum_{i=0}^{M}\sum_{j=0}^{N}a_{ij}P_{\tau,i}^{(\rho_2,\gamma_2)}(y)P_{L,j}^{(\rho_1,\gamma_1)}(z)dydz$$
$$+g(\varrho,\sigma) \qquad (4.3)$$

In the introduced SJ-G-C technique, the remaining of (4.3) is set to be zero at $(N + 1) \times (M + 1)$ of SJ-G points

$$\sum_{i=0}^{M}\sum_{j=0}^{N}a_{ij}\eta_{i,j}^{l,m} = g(\varrho_{L,N,l}^{(\rho_1,\gamma_1)}, s_{L,N,m}^{(\rho_2,\gamma_2)}), l = 0,\dots,N, \ m = 0,\dots,M, \qquad (4.4)$$

where

$$\eta_{i,j}^{l,m} = -\int_{0}^{\varrho_{L,N,l}^{(\rho_1,\gamma_1)}}\int_{0}^{L}K(\varrho_{L,N,l}^{(\rho_1,\gamma_1)}, \sigma_{L,N,m}^{(\rho_2,\gamma_2)}, y, z)\sum_{i=0}^{M}\sum_{j=0}^{N}a_{ij}P_{\tau,i}^{(\rho_2,\gamma_2)}(y)P_{L,j}^{(\rho_1,\gamma_1)}(z)dydz$$
$$+P_{\tau,i}^{(\rho_2,\gamma_2)}(\sigma_{L,N,m}^{(\rho_2,\gamma_2)})P_{L,j}^{(\rho_1,\gamma_1)}(\varrho_{L,N,l}^{(\rho_1,\gamma_1)}).$$

Lastly, Eq (4.4) guarantees that the equation (4.1) is satisfied exactly at the SJ-G interpolation nodes $\varrho_{L,N,l}^{(\rho_1,\gamma_1)}, \sigma_{L,N,m}^{(\rho_2,\gamma_2)}; \ l = 0,\cdots,N,; \ m = 0,\cdots,M$. This provides $(N + 1) \times (M + 1)$ algebraic equations for $a_{i,j}; i = 0,\cdots,N, j = 0,\cdots,M$. Hence, the convergent solutions (4.2) can be evaluated model

$$\Lambda_{N,M}(\varrho,\sigma) = \sum_{i=0}^{M}\sum_{j=0}^{N}a_{ij}P_{L,j}^{(\rho_1,\gamma_1)}(\varrho)P_{\tau,i}^{(\rho_2,\gamma_2)}(\sigma). \qquad (4.5)$$

## 4.2 Non-linear mixed Volterra-Fredholm integral equations

In this subsection, we enlarge the preceding technique for the numerical treating of the nonlinear next form,

$$\Lambda(\varrho,\sigma) = \int_{0}^{t}\int_{0}^{L}K(\varrho,\sigma,y,z,\Lambda(y,z))dydz, +g(\varrho,\sigma) \qquad 0 \le \varrho,\sigma \le L, \qquad (4.6)$$

wherever $g(\varrho,\sigma)$ and $K(\varrho,\sigma,y,z)$ are given functions, whilst $\Lambda(\varrho,\sigma)$ is an unknowna function.

As a consequence of the methodology presented in the preceding subsection, we can write the problem in the following model

$$\sum_{i=0}^{M}\sum_{j=0}^{N}a_{ij}P_{L,j}^{(\rho_1,\gamma_1)}(\varrho)P_{\tau,i}^{(\rho_2,\gamma_2)}(\sigma) = \int_{0}^{t}\int_{0}^{L}K(\varrho,\sigma,y,z,\sum_{i=0}^{M}\sum_{j=0}^{N}a_{ij}P_{\tau,i}^{(\rho_2,\gamma_2)}(y)P_{L,j}^{(\rho_1,\gamma_1)}(z))dydz$$
$$+g(\varrho,\sigma). \qquad (4.7)$$

The residual of (4.6) is set to be zero at $N + 1$ of SJ-G points given the next model:

$$\sum_{i=0}^{M}\sum_{j=0}^{N} a_{ij} P_{\tau,i}^{(\rho_2,\gamma_2)}(\sigma_{L,N,m}^{(\rho_2,\gamma_2)}) P_{L,j}^{(\rho_1,\gamma_1)}(\varrho_{L,N,l}^{(\rho_1,\gamma_1)}) = \int_{0}^{\varrho_{L,N,l}^{(\rho_1,\gamma_1)}} \int_{0}^{L} \chi_{l,m}(y,z) dy dz + g(\varrho_{L,N,l}^{(\rho_1,\gamma_1)}, s_{L,N,m}^{(\rho_2,\gamma_2)}), \qquad (4.8)$$

$$l = 0,\ldots,N,\ m = 0,\ldots,M,$$

wherever

$$\chi_{l,m}(y,z) = K(\varrho_{L,N,l}^{(\rho_1,\gamma_1)}, s_{L,N,m}^{(\rho_2,\gamma_2)}, y, z, \sum_{i=0}^{M}\sum_{j=0}^{N} a_{ij} P_{\tau,i}^{(\rho_2,\gamma_2)}(y) P_{L,j}^{(\rho_1,\gamma_1)}(z)).$$

It can be solved the preceding nonlinear algebraic system, for $a_{ij}$, by utilized Newton's iterative method for counting the approximate solution $\Lambda_{N,M}(\varrho, \sigma)$ at any value of $(\varrho, \sigma)$ in the specified domain, by means of the next equation

$$\Lambda_{N,M}(\varrho, \sigma) = \sum_{i=0}^{M}\sum_{j=0}^{N} a_{ij} P_{L,j}^{(\rho_1,\gamma_1)}(\varrho) P_{\tau,i}^{(\rho_2,\gamma_2)}(\sigma). \qquad (4.9)$$

## 5 Some useful lemmas and error analysis

In this section, we restate some useful lemmas [31, 32] and discuss the convergence analysis of the technique introduced for the linear MV-FIEs.

### 5.1 Some useful lemmas

Let $P_N: L^2(I) \to Y_N$ be the $L^2$ orthogonal projection, defined by [33–35]

$$(P_N\Lambda - \Lambda, v) = 0, \forall v \in Y_N.$$

We first present several weighted Hilbert spaces. For simplicity, denote $\partial_y v(y) = (\partial/\partial y)v(y)$, etc. For a nonnegative integer $m$, we realize [35–37]

$$H_{w^{\rho,\gamma}}^m(-1,1) = \{v : \partial_y^i v \in L_{w^{\rho,\gamma}}^2(-1,1), 0 \le i \le m\},$$

with the semi-norm and the norm as

$$v_{m,w^{\rho,\gamma}} = \|\partial_y^m v\|_{w^{\rho,\gamma}},$$

$$\|v\|_{m,w^{\rho,\gamma}} = (\sum_{i=0}^{m}\|\partial_y^i v\|_{w^{\rho,\gamma}}^2)^{\frac{1}{2}}.$$

Assume that $\Lambda \in H^m(I)$ and denoted $I_N^{\rho,\gamma}$ the interpolation of $\Lambda$ at any of the three families of Jacobi Gauss points (Gauss or Gauss-Radau or Gauss-lobatto). Then for I = (-1,1) the next estimate holds [36]:

$$\|\Lambda - I_N^{\rho,\gamma}\Lambda\|_{L_{w^{\rho,\gamma}}^2(I)} \le CN^{-m} \mid \Lambda \mid_{H_{w^{\rho,\gamma}}^{m,N}(I)}. \qquad (5.1)$$

Let $I_N^{\rho,\gamma}\Lambda \in P_N$ denoted the interpolation of $\Lambda$ at any of the three families of Gauss points (Gauss or Gauss-Radau or Gauss-lobatto), if $\Lambda \in H^m(I)$. Then for I = (-1,1) the following

estimates hold: (see [36] pages 297 and 310)

$$\|\Lambda - I_N^{\rho,\gamma}\Lambda\|_{L_\infty(I)} \leq CN^{\frac{1}{2}-m} \mid \Lambda\mid_{H_{w^{\rho,\gamma}}^{m,N}(I)}. \tag{5.2}$$

## 5.2 Error analysis

The major objective is to assess the spectral convergence of the introduced technique. In the subsection, error analysis for the numerical schemes for linear MV-F-IEs (3.1) will be provided, which indicates that the numerical errors decay exponentially provided that the kernel function and the source function are sufficiently smooth. Assume $\Lambda(y)$ be exact solution of linear MV-F-IEs (3.1) and suppose that $I_N^{\rho,\gamma}(\Lambda(y)) = \Lambda_N(y)$ be the spectral collection approximation specified by equation (3.5) therefore sufficiently smooth function $g(y)$, $k_1(y, t)$ and $k_2(y, t)$ in (3.1) and for all sufficiently large $N$ we have

$$\begin{aligned}
\|\Lambda(y) - \Lambda_N(y)\|_{L_{w^{\rho,\gamma}}^2(I)} \leq \quad & N^{-m}\big(C_4 \mid k_1(\varrho,\sigma)\mid_{H_{w^{\rho,\gamma}}^{m,N}(I)}\|\Lambda\|_{L_{w^{\rho,\gamma}}^2(I)} + \gamma C_5 N^{\frac{1}{2}} \mid \Lambda\mid_{H_{w^{\rho,\gamma}}^{m,N}(I)} \\
& +C_8 \mid k_2(\varrho,\sigma)\mid_{H_{w^{\rho,\gamma}}^{m,N}(I)}\|\Lambda\|_{L_{w^{\rho,\gamma}}^2(I)} + \|k_2\|_\infty C_9 N^{\frac{1}{2}} \mid \Lambda\mid_{H_{w^{\rho,\gamma}}^{m,N}(I)}\big).
\end{aligned} \tag{5.3}$$

**Proof**

Assume the equation of MV-F-IEs

$$\Lambda(\varrho) = g(\varrho) + \int_0^{h(\varrho)} k_1(\varrho,\sigma)\Lambda(\sigma)d\sigma + \int_0^L k_2(\varrho,\sigma)\Lambda(\sigma)d\sigma, \tag{5.4}$$

and define the error function

$$e_N(\Lambda(\varrho)) = \Lambda(\varrho) - I_N^{\rho,\gamma}(\Lambda(\varrho)), e_{N,N}(k(\varrho,\sigma)) = k(\varrho,\sigma) - I_{N,N}^{\rho,\gamma}(k(\varrho,\sigma)),$$

wherever $\Lambda(y)$ is a continuous function, then

$$\Lambda_N(\varrho) = g(y) + \int_0^{h(\varrho)} I_{N,N}^{\rho,\gamma}(k_1(\varrho,\sigma))I_N^{\rho,\gamma}\Lambda(\sigma)d\sigma + \int_0^L I_{N,N}^{\rho,\gamma}(k_2(\varrho,\sigma))I_N^{\rho,\gamma}\Lambda(\sigma)d\sigma, \tag{5.5}$$

subtracting (5.5) from (5.4), we get

$$\begin{aligned}
e_N(\varrho) = \int_0^{h(\varrho)} k_1(\varrho,\sigma)\Lambda(\sigma)d\sigma \quad & -\int_0^{h(\varrho)} I_{N,N}^{\rho,\gamma}k_1(\varrho,\sigma)I_N^{\rho,\gamma}\Lambda(\sigma)d\sigma + \int_0^L (k_2(\varrho,\sigma))\Lambda(\sigma)d\sigma \\
& -\int_0^L I_{N,N}^{\rho,\gamma}(k_2(\varrho,\sigma))I_N^{\rho,\gamma}\Lambda(\sigma)d\sigma,
\end{aligned} \tag{5.6}$$

$$e_N(\varrho) = \int_0^{h(\varrho)} e_{N,N}(k_1(\varrho,\sigma))I_N^{\rho,\gamma}\Lambda(\sigma)d\sigma + \int_0^{h(\varrho)} (k_1(\varrho,\sigma))\Lambda(\sigma)d\sigma - \int_0^{h(\varrho)} (k_1(\varrho,\sigma))I_N^{\rho,\gamma}\Lambda(\sigma)d\sigma$$

$$+ \int_0^{L} e_{N,N}(k_2(\varrho,\sigma))I_N^{\rho,\gamma}\Lambda(\sigma)d\sigma + \int_0^{L} (k_2(\varrho,\sigma))\Lambda(\sigma)d\sigma - \int_0^{L} (k_2(\varrho,\sigma))I_N^{\rho,\gamma}\Lambda(\sigma)d\sigma,$$

(5.7)

$$e_N(\varrho) = \int_0^{h(\varrho)} e_{N,N}(k_1(\varrho,\sigma))I_N^{\rho,\gamma}\Lambda(\sigma)d\sigma + \int_0^{h(\varrho)} (k_1(\varrho,\sigma))(\Lambda(\sigma) - I_N^{\rho,\gamma}\Lambda(\sigma))d\sigma$$

$$+ \int_0^{L} e_{N,N}(k_2(\varrho,\sigma))I_N^{\rho,\gamma}\Lambda(\sigma)d\sigma + \int_0^{L} (k_2(\varrho,\sigma))(\Lambda(\sigma) - I_N^{\rho,\gamma}\Lambda(\sigma))d\sigma.$$

(5.8)

Then

$$e_N(y) = L_1 + L_2 + L_3 + L_4,$$

where

$$L_1 = \int_0^{h(\varrho)} e_{N,N}(k_1(\varrho,\sigma))I_N^{\rho,\gamma}\Lambda(\sigma)d\sigma , L_2 = \int_0^{h(\varrho)} k_1(\varrho,\sigma)(\Lambda(\sigma) - I_N^{\rho,\gamma}\Lambda(\sigma))d\sigma,$$

$$L_3 = \int_0^{L} e_{N,N}k_2(\varrho,\sigma)I_N^{\rho,\gamma}\Lambda(\sigma)d\sigma, L_4 = \int_0^{L} k_2(\varrho,\sigma)(\Lambda(\sigma) - I_N^{\rho,\gamma}\Lambda(\sigma))d\sigma.$$

From Grönwall inequality we write

$$\|e_N(\varrho)\|_{L^2_{w^{\rho,\gamma}}(I)} \leq \|L_1\|_{L^2_{w^{\rho,\gamma}}(I)} + \|L_2\|_{L^2_{w^{\rho,\gamma}}(I)} + \|L_3\|_{L^2_{w^{\rho,\gamma}}(I)} + \|L_4\|_{L^2_{w^{\rho,\gamma}}(I)}.$$

(5.9)

Utilizing Cauchy-Schwartz inequality, we can write

$$\|L_1\|_{L^2_{w^{\rho,\gamma}}(I)} \leq \|e_{N,N}k_1(\varrho,\sigma)\|_{L^2_{w^{\rho,\gamma}}(I)} \|I_N^{\rho,\gamma}\Lambda(\sigma)\|_{L^2_{w^{\rho,\gamma}}(I)}.$$

(5.10)

Now from [36, 38], we have

$$\|e_{N,N}k_1(\varrho,\sigma)\|_{L^2_{w^{\rho,\gamma}}(I)} \leq \|k_1(\varrho,\sigma) - I_{N,N}^{\rho,\gamma}k_1(\varrho,\sigma)\|_{L^2_{w^{\rho,\gamma}}(I)} \leq C_1 N^{-m} \mid k_1(\varrho,\sigma)\mid_{H^{m,N}_{w^{\rho,\gamma}}(I)},$$

(5.11)

an upper bound for (5.10) follows from (5.11) as

$$\|L_1\|_{L^2_{w^{\rho,\gamma}}(I)} \leq (C_2 N^{-m} \mid k_1(\varrho,\sigma)\mid_{H^{m,N}_{w^{\rho,\gamma}}(I)})(C_3\|\Lambda\|_{L^2_{w^{\rho,\gamma}}(I)}).$$

(5.12)

Then

$$\|L_1\|_{L^2_{w^{\rho,\gamma}}(I)} \leq C_4 N^{-m} \mid k_1(\varrho,\sigma)\mid_{H^{m,N}_{w^{\rho,\gamma}}(I)}\|\Lambda\|_{L^2_{w^{\rho,\gamma}}(I)}.$$

(5.13)

From [39], yields

$$
\begin{aligned}
\|L_2\|_{L_{w^{\rho,\gamma}}^2(I)} &= \|\int_0^{h(\varrho)} k_1(\varrho,\sigma)(\Lambda(\sigma) - I_N^{\rho,\gamma}\Lambda(\sigma))d\sigma\|_{L_{w^{\rho,\gamma}}^2(I)} \\
&\leq \|\int_0^{h(\varrho)} k_1(\varrho,\sigma)(\Lambda(\sigma) - I_N^{\rho,\gamma}\Lambda(\sigma))d\sigma\|_{\infty} \\
&\leq \gamma\|(\Lambda(\sigma) - I_N^{\rho,\gamma}\Lambda(\sigma))\|_{\infty},
\end{aligned}
\tag{5.14}
$$

where

$$
\gamma = \max_{a \leq h(\varrho) \leq b} \mid k_1(\varrho,\sigma) \mid .
$$

Therefore from Lemma 5.1, we find

$$
\|L_2\|_{L_{w^{\rho,\gamma}}^2(I)} \leq \gamma C_5 N^{\frac{1}{2}-m} \mid \Lambda|_{H_{w^{\rho,\gamma}}^{m,N}(I)}.
\tag{5.15}
$$

Using Cauchy-Schwartz inequality, we write

$$
\|L_3\|_{L_{w^{\rho,\gamma}}^2(I)} \leq \|e_{N,N}k_2(\varrho,\sigma)\|_{L_{w^{\rho,\gamma}}^2(I)}\|I_N^{\rho,\gamma}\Lambda(\sigma)\|_{L_{w^{\rho,\gamma}}^2(I)}.
\tag{5.16}
$$

From Eq (5.11), we obtain

$$
\|L_3\|_{L_{w^{\rho,\gamma}}^2(I)} \leq (C_6 N^{-m} \mid k_2(\varrho,\sigma)|_{H_{w^{\rho,\gamma}}^{m,N}(I)})(C_7\|\Lambda\|_{L_{w^{\rho,\gamma}}^2(I)}).
\tag{5.17}
$$

Then

$$
\|L_3\|_{L_{w^{\rho,\gamma}}^2(I)} \leq C_8 N^{-m} \mid k_2(\varrho,\sigma)|_{H_{w^{\rho,\gamma}}^{m,N}(I)}\|\Lambda\|_{L_{w^{\rho,\gamma}}^2(I)},
\tag{5.18}
$$

$$
\begin{aligned}
\|L_4\|_{L_{w^{\rho,\gamma}}^2(I)} &= \|\int_0^{L} k_2(\varrho,\sigma)(\Lambda(\varrho) - I_N^{\rho,\gamma}\Lambda(\sigma))d\sigma\|_{L_{w^{\rho,\gamma}}^2(I)} \\
&\leq \|\int_0^{L} k_2(\varrho,\sigma)(\Lambda(\sigma) - I_N^{\rho,\gamma}\Lambda(\sigma))d\sigma\|_{\infty} \\
&\leq \|k_2\|_{\infty}\|(\Lambda(\sigma) - I_N^{\rho,\gamma}\Lambda(\sigma))\|_{\infty},
\end{aligned}
\tag{5.19}
$$

where [39]

$$
\|k_2\|_{\infty} = \max \int_0^{L} \mid k_2(\varrho,\sigma) \mid d\sigma < \infty.
$$

Therefore from Lemma 5.1, we obtain

$$
\|L_4\|_{L_{w^{\rho,\gamma}}^2(I)} \leq \|k_2\|_{\infty} C_9 N^{\frac{1}{2}-m} \mid \Lambda|_{H_{w^{\rho,\gamma}}^{m,N}(I)}.
\tag{5.20}
$$

Using Eqs (5.9), (5.13), (5.15), (5.18) and (5.20), we obtain

$$
\|\Lambda(y) - \Lambda_N(y)\quad\|_{L^2_{w^{\rho,\gamma}}(I)} \leq C_4 N^{-m} \mid k_1(\varrho, \sigma)\mid_{H^{m,N}_{w^{\rho,\gamma}}(I)}\|\Lambda\|_{L^2_{w^{\rho,\gamma}}(I)} + \gamma C_5 N^{\frac{1}{2}-m} \mid \Lambda\mid_{H^{m,N}_{w^{\rho,\gamma}}(I)}
$$
$$
+ C_8 N^{-m} \mid k_2(\varrho, \sigma)\mid_{H^{m,N}_{w^{\rho,\gamma}}(I)}\|\Lambda\|_{L^2_{w^{\rho,\gamma}}(I)} + \|k_2\|_\infty C_9 N^{\frac{1}{2}-m} \mid \Lambda\mid_{H^{m,N}_{w^{\rho,\gamma}}(I)}.
\tag{5.21}
$$

Then

$$
\|\Lambda(y) - \Lambda_N(y)\quad\|_{L^2_{w^{\rho,\gamma}}(I)} \leq N^{-m}(C_4 \mid k_1(\varrho, \sigma)\mid_{H^{m,N}_{w^{\rho,\gamma}}(I)}\|\Lambda\|_{L^2_{w^{\rho,\gamma}}(I)} + \gamma C_5 N^{\frac{1}{2}} \mid \Lambda\mid_{H^{m,N}_{w^{\rho,\gamma}}(I)}
$$
$$
+ C_8 \mid k_2(\varrho, \sigma)\mid_{H^{m,N}_{w^{\rho,\gamma}}(I)}\|\Lambda\|_{L^2_{w^{\rho,\gamma}}(I)} + \|k_2\|_\infty C_9 N^{\frac{1}{2}} \mid \Lambda\mid_{H^{m,N}_{w^{\rho,\gamma}}(I)}).
\tag{5.22}
$$

the proof of the theorem is concluded.

## 6 Numerical results

To display the performance and accuracy of our scheme, we adapt some numerical examples. We also give comparisons between our results with other methods [12–14, 16]. In each of these examples, we shall highlight the accuracy and convergence of the proposed technique. The programs used in this work are run on a PC with an Intel(R) Core(TM) i7-10510U CPU running @ 1.80GHz and 2.30GHz, 2.00 GB of RAM, and Mathematica version 12 running the code.

The absolute error (AEs) is defined as:

$$
E(\varrho) = \mid \Lambda(\varrho) - \Lambda_N(\varrho) \mid,
\tag{6.1}
$$

where $\Lambda(\varrho)$ and $\Lambda_N(\varrho)$ are the approximate and exact solutions at $\varrho$.

To assess the results, we use the maximum absolute error (MAEs), defined as:

$$
\text{MAEs} = \text{Max}\{|E(\varrho)|\}.
\tag{6.2}
$$

### 6.1 One-dimensional mixed Volterra-Fredholm integral equations

**Example 1** *Assume the next one-dimensional linear MV-F-IEs* [16]

$$
\Lambda(\varrho) = g(\varrho) + \int_0^\varrho e^\sigma \cos(\varrho)\Lambda(\sigma)d\sigma - \int_0^1 e^\sigma \sin(\varrho)\Lambda(\sigma)d\sigma, \varrho \in [0, 1],
\tag{6.3}
$$

*where $g(\varrho) = e^\varrho - \frac{1}{2}\cos(\varrho)(e^{2\varrho} - 1) + \frac{1}{2}\sin(\varrho)(e^2 - 1)$. The exact solution is $\Lambda(\varrho) = e^\varrho$.*

To confirm the high accuracy of our introduced technique, we list the MAEs for the preceding problem in Table 1 with various option of $N, \rho, \gamma$. Table 1 presents a comparison between our method and Lagrange collocation method (LCM) [16] for $N = 13$. This comparison display that, we have numerical solutions of bestead thoroughness with far fewer nodes. Also, we observe that our numerical solutions correspond carefully to the exact ones.

In Figs 1 and 2, we see the matching of the AEs values in this figure and that in Table 1. Whilst, Fig 3 compare graphically between the curves of exact and numerical solutions. In Fig 4, we introduced the logarithmic graphs of $M_E$ ($log_{10} M_E$) acquired by the current technique with various value of $\rho, \gamma$, and $N$. The acquired results emphasis the high thoroughness and exponential convergence of the current scheme, which demonstrated the theoretical analysis of the convergence.

**Table 1. MAEs for Example 1.**

| N | New Method | | | LCM [16] |
|---|---|---|---|---|
| | $\rho = \gamma = -\frac{1}{2}$ | $\rho = \gamma = 0$ | $\rho = 0, \gamma = \frac{1}{2}$ | |
| 2 | $3.1028 \times 10^{-2}$ | $1.4552 \times 10^{-2}$ | $2.4548 \times 10^{-2}$ | $1.0589 \times 10^{-2}$ |
| 5 | $1.1875 \times 10^{-6}$ | $2.5499 \times 10^{-6}$ | $5.5480 \times 10^{-6}$ | $1.1211 \times 10^{-6}$ |
| 8 | $3.5906 \times 10^{-11}$ | $9.5239 \times 10^{-10}$ | $2.4525 \times 10^{-11}$ | $9.3486 \times 10^{-7}$ |
| 13 | $4.4409 \times 10^{-16}$ | $4.0453 \times 10^{-16}$ | $4.4409 \times 10^{-16}$ | – |

**Example 2** *Assume us consider the next MV-F-IEs* [12]

$$\varrho^2 \Lambda(\varrho) + e^\varrho \Lambda(\varrho) = g(\varrho) + \int_0^{h(\varrho)} e^{\varrho+\sigma} \Lambda(\sigma)d\sigma - \int_0^1 e^{\varrho-h(\sigma)} \Lambda(h(\sigma))d\sigma, \tag{6.4}$$

*where*

$$g(\varrho) = -\frac{e^\varrho}{4} - \frac{1}{4}e^{\varrho-2}\cos(2) + \frac{1}{2}e^{3\varrho}\cos(2\varrho) - \frac{1}{4}e^{\varrho-2}\sin(2) + \varrho^2\sin(\varrho) + e^\varrho\sin(2\varrho) - \frac{1}{2}e^{3\varrho}\sin(2\varrho),$$

$$h(\varrho) = 2\varrho,$$

*and its exact solution given by*

$$\Lambda(\varrho) = \sin(\varrho).$$

A comparison between the MAEs acquired by utilizing the proposed technique and Legender collocation way (L-CM) [12], Taylor collocation method (TCM) [15], Taylor polynomial method (TPM) [10] and LCM [16] is abbreviated in Table 2 with several choices of $\rho$ and $\gamma$.

**Example 3** *We examine the next MV-F-IEs with the exact solution* $\Lambda(\varrho) = e^{-\varrho}$ [12]

$$\Lambda(\varrho) = g(\varrho) + \int_0^{h(\varrho)} e^{\varrho+\sigma} \Lambda(\sigma)d\sigma - \int_0^1 e^{\varrho+h(\sigma)} \Lambda(h(\sigma))d\sigma, \tag{6.5}$$

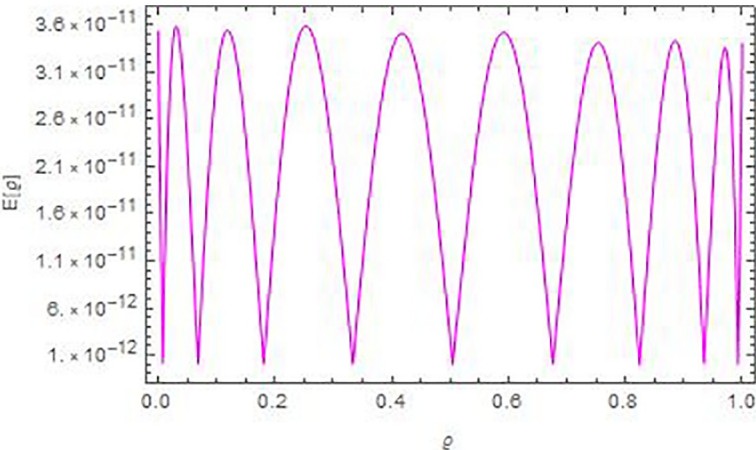

**Fig 1. The AEs curve versus $\varrho$ in Example 1, for $N = 8$ and $\rho = \gamma = -\frac{1}{2}$.**

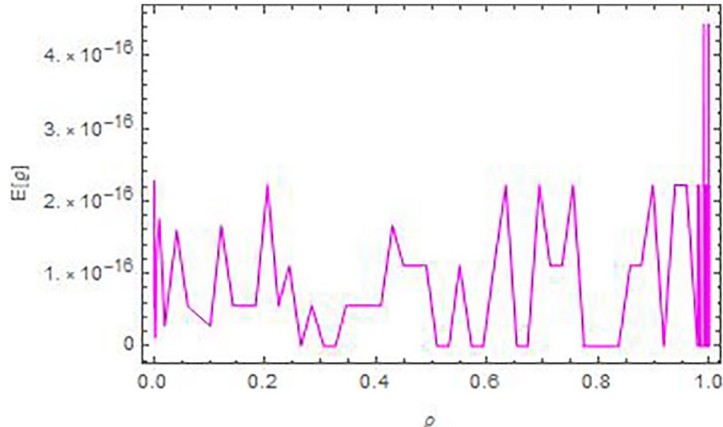

**Fig 2. The AEs curve versus $\varrho$ in Example 1, for $N = 13$ and $\rho = \gamma = -\frac{1}{2}$.**

*where*

$$g(\varrho) = e^{-\varrho} - e^{\varrho}(h(\varrho) - 1), h(\varrho) = \ln(\varrho + 1).$$

A comparison between the MAEs acquired by utilizing the proposed technique with SCC [40], TCM [15], TPM [10] and LCM [16] is listed in Table 3 with several choices of $\rho$ and $\gamma$.

**Example 4** *We test the next non-linear MV-F-IEs with the exact solution* $\Lambda(\varrho) = \varrho^2 \sin(\varrho)$ [12]

$$\Lambda(\varrho) = g(\varrho) + \frac{1}{4}\int_0^{\varrho}(\varrho - \sigma)(\Lambda(\sigma))^2 d\sigma + \int_0^1(1 + \sigma)\Lambda(\sigma)d\sigma, \tag{6.6}$$

*where*

$$g(\varrho) = \frac{143}{64} - \frac{\varrho^6}{240} - \frac{1}{64}(15 - 18\varrho^2 + 2\varrho^4)\cos(2\varrho) + \frac{1}{4}\varrho(4\varrho + (-3 + \varrho^2)\cos(\varrho))\sin(\varrho) - 6\cos(1) + \sin(1).$$

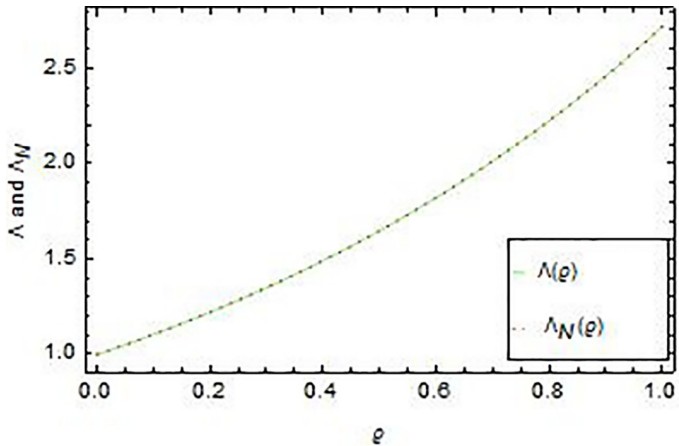

**Fig 3. Comparison between the exact $\Lambda(\varrho)$ and the approximate $\Lambda_N(\varrho)$ solutions in Example 1 for $N = 13$ and $\rho = -\frac{1}{2}, \gamma = \frac{1}{2}$.**

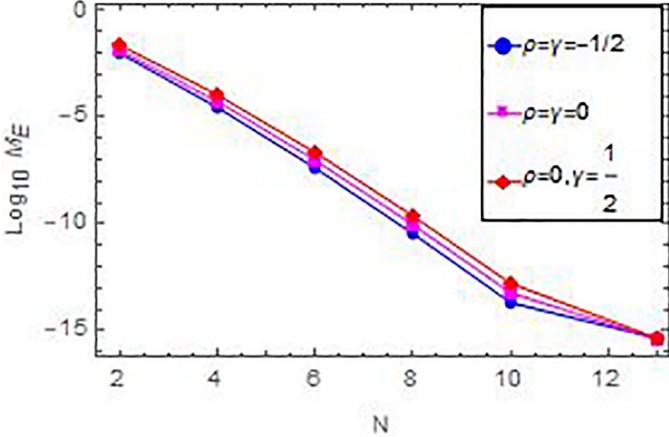

**Fig 4. Convergence for Example 1.**

A comparison between the MAEs acquired by utilizing the proposed technique and least square way [11], is abbreviated in Table 4 with several choices of $N$.

## 6.2 Two-dimensional mixed Volterra-Fredholm integral equations

**Example 5** *Consider the following MV-F-IEs* [8, 13]

$$\Lambda(\varrho, \sigma) = g(\varrho, \sigma) - \int\limits_{0}^{\varrho} \int\limits_{0}^{1} \sigma^2 e^{-z} \Lambda(y, z) dy dz, \qquad (6.7)$$

*where $g(\varrho, \sigma) = \frac{1}{3} \varrho^2 (3e^{\sigma} + 3\varrho\sigma^2)$. The exact solution is $\Lambda(\varrho, \sigma) = \varrho^2 e^{\sigma}$*

In the Table 5, a comparison between the AEs acquired in [13] and the results acquired in this paper with various values of $N$ and $M$ of Example 5 is displayed. Furthermore, Table 6 summarizes some value of MAEs with various values of $N$ and $M$.

**Table 2. MAEs for Example 2.**

| N | Present Method | | | |
|---|---|---|---|---|
| | $\rho = \gamma = -\frac{1}{2}$ | $\rho = \gamma = 0$ | $\rho = 0, \gamma = \frac{1}{2}$ | |
| 2 | $3.413 \times 10^{-2}$ | $4.867 \times 10^{-2}$ | $1.572 \times 10^{-1}$ | |
| 5 | $5.726 \times 10^{-5}$ | $9.983 \times 10^{-5}$ | $2.020 \times 10^{-4}$ | |
| 8 | $8.063 \times 10^{-9}$ | $2.108 \times 10^{-8}$ | $5.167 \times 10^{-8}$ | |
| 9 | $5.552 \times 10^{-10}$ | $1.428 \times 10^{-9}$ | $3.834 \times 10^{-9}$ | |
| 12 | $2.647 \times 10^{-14}$ | $8.481 \times 10^{-14}$ | $2.473 \times 10^{-13}$ | |
| | L-CM [12] | TCM [15] | TPM [10] | LCM [16] |
| 2 | $1.46 \times 10^{-2}$ | $7.87 \times 10^{-2}$ | $3.41 \times 10^{-2}$ | $7.87 \times 10^{-2}$ |
| 5 | $2.93 \times 10^{-5}$ | $6.23 \times 10^{-5}$ | $3.68 \times 10^{-4}$ | $6.23 \times 10^{-5}$ |
| 8 | $3.94 \times 10^{-8}$ | $1.89 \times 10^{-8}$ | $1.24 \times 10^{-5}$ | $1.77 \times 10^{-7}$ |
| 9 | $2.29 \times 10^{-9}$ | $2.35 \times 10^{-8}$ | $3.46 \times 10^{-7}$ | $7.21 \times 10^{-6}$ |

**Table 3. MAEs for Example 3.**

| N | Our Method | | | |
|---|---|---|---|---|
| | $\rho = \gamma = -\frac{1}{2}$ | $\rho = \gamma = 0$ | $\rho = 0, \gamma = \frac{1}{2}$ | |
| 2 | $4.70 \times 10^{-3}$ | $6.69 \times 10^{-3}$ | $1.02 \times 10^{-2}$ | |
| 5 | $4.80 \times 10^{-7}$ | $1.04 \times 10^{-6}$ | $2.17 \times 10^{-6}$ | |
| 8 | $1.52 \times 10^{-11}$ | $3.76 \times 10^{-11}$ | $9.39 \times 10^{-11}$ | |
| 9 | $4.17 \times 10^{-13}$ | $9.73 \times 10^{-13}$ | $2.56 \times 10^{-12}$ | |
| 12 | $4.44 \times 10^{-16}$ | $2.91 \times 10^{-16}$ | $3.33 \times 10^{-16}$ | |
| N | SCC [40] | TCM [15] | TPM [10] | LCM [16] |
| 2 | $4.78 \times 10^{-3}$ | $3.27 \times 10^{-3}$ | $3.59 \times 10^{-2}$ | $3.27 \times 10^{-3}$ |
| 5 | $4.83 \times 10^{-7}$ | $4.30 \times 10^{-7}$ | $3.05 \times 10^{-4}$ | $4.30 \times 10^{-7}$ |
| 8 | $1.43 \times 10^{-11}$ | $5.96 \times 10^{-8}$ | $5.61 \times 10^{-7}$ | $5.78 \times 10^{-7}$ |
| 9 | $4.24 \times 10^{-13}$ | $8.84 \times 10^{-8}$ | $1.41 \times 10^{-7}$ | $1.86 \times 10^{-5}$ |

**Table 4. MAEs for Example 4.**

| N | Least square method [11] | | | Present Method |
|---|---|---|---|---|
| | $M = 1$ | $M = 2$ | $M = 3$ | $N = M$ |
| 5 | $2.6 \times 10^{-3}$ | $1.6 \times 10^{-3}$ | $9 \times 10^{-4}$ | $1.5 \times 10^{-5}$ |
| 9 | $1.6 \times 10^{-3}$ | $4.7 \times 10^{-4}$ | $1.4 \times 10^{-4}$ | $3.0 \times 10^{-11}$ |
| 13 | $1.2 \times 10^{-3}$ | $2 \times 10^{-4}$ | $2.5 \times 10^{-5}$ | $4.6 \times 10^{-16}$ |
| 23 | $7 \times 10^{-4}$ | $5 \times 10^{-5}$ | $4 \times 10^{-6}$ | – |

**Table 5. AEs for Example 5.**

| $(\varrho_i, s_i)$ | MQs L-G-L [13] | | | |
|---|---|---|---|---|
| | $N = M = 2$ | $N = M = 3$ | $N = M = 4$ | $N = M = 5$ |
| (0,0) | $5.22 \times 10^{-4}$ | $4.58 \times 10^{-4}$ | $4.69 \times 10^{-4}$ | $1.64 \times 10^{-6}$ |
| (0.1,0.1) | $1.01 \times 10^{-5}$ | $3.44 \times 10^{-4}$ | $2.55 \times 10^{-5}$ | $7.68 \times 10^{-5}$ |
| (0.2,0.2) | $7.86 \times 10^{-4}$ | $9.99 \times 10^{-4}$ | $3.32 \times 10^{-4}$ | $4.19 \times 10^{-4}$ |
| (0.3,0.3) | $4.69 \times 10^{-3}$ | $1.79 \times 10^{-3}$ | $2.36 \times 10^{-3}$ | $2.71 \times 10^{-3}$ |
| (0.4,0.4) | $1.37 \times 10^{-2}$ | $1.22 \times 10^{-2}$ | $1.00 \times 10^{-2}$ | $9.94 \times 10^{-3}$ |
| (0.5,0.5) | $3.05 \times 10^{-2}$ | $3.52 \times 10^{-2}$ | $3.06 \times 10^{-2}$ | $2.99 \times 10^{-2}$ |
| (0.6,0.6) | $6.45 \times 10^{-2}$ | $7.93 \times 10^{-2}$ | $7.61 \times 10^{-2}$ | $7.57 \times 10^{-2}$ |
| (0.7,0.7) | $1.39 \times 10^{-01}$ | $1.61 \times 10^{-01}$ | $1.64 \times 10^{-01}$ | $1.64 \times 10^{-01}$ |
| (0.8,0.8) | $2.93 \times 10^{-01}$ | $3.08 \times 10^{-01}$ | $3.17 \times 10^{-01}$ | $3.17 \times 10^{-01}$ |
| (0.9,0.9) | $5.77 \times 10^{-01}$ | $5.65 \times 10^{-01}$ | $5.70 \times 10^{-01}$ | $5.69 \times 10^{-01}$ |
| | Present Method | | | |
| $(\varrho_i, s_i)$ | $N = M = 2$ | $N = M = 4$ | $N = M = 8$ | $N = M = 12$ |
| (0,0) | $0$ | $9.17 \times 10^{-17}$ | $4.16 \times 10^{-17}$ | $1.13 \times 10^{-16}$ |
| (0.1,0.1) | $1.01 \times 10^{-5}$ | $2.05 \times 10^{-7}$ | $1.70 \times 10^{-13}$ | $8.67 \times 10^{-18}$ |
| (0.2,0.2) | $1.85 \times 10^{-4}$ | $3.19 \times 10^{-7}$ | $1.68 \times 10^{-13}$ | $2.08 \times 10^{-17}$ |
| (0.3,0.3) | $5.22 \times 10^{-4}$ | $1.29 \times 10^{-6}$ | $1.56 \times 10^{-12}$ | $2.50 \times 10^{-16}$ |
| (0.4,0.4) | $6.05 \times 10^{-4}$ | $2.66 \times 10^{-6}$ | $3.66 \times 10^{-15}$ | $3.19 \times 10^{-16}$ |
| (0.5,0.5) | $5.21 \times 10^{-8}$ | $5.17 \times 10^{-13}$ | $5.55 \times 10^{-17}$ | $5.55 \times 10^{-17}$ |
| (0.6,0.6) | $1.43 \times 10^{-3}$ | $6.18 \times 10^{-6}$ | $8.40 \times 10^{-12}$ | $1.11 \times 10^{-16}$ |
| (0.7,0.7) | $3.14 \times 10^{-3}$ | $7.53 \times 10^{-6}$ | $8.86 \times 10^{-12}$ | $1.11 \times 10^{-16}$ |
| (0.8,0.8) | $3.44 \times 10^{-3}$ | $5.64 \times 10^{-6}$ | $2.86 \times 10^{-12}$ | $0$ |
| (0.9,0.9) | $9.95 \times 10^{-4}$ | $1.90 \times 10^{-5}$ | $1.49 \times 10^{-11}$ | $0$ |

**Table 6. MAEs for Example 5.**

| N = M | 2 | 4 | 8 | 12 |
|---|---|---|---|---|
| MAE | $1.576 \times 10^{-2}$ | $5.980 \times 10^{-5}$ | $9.881 \times 10^{-11}$ | $1.332 \times 10^{-15}$ |

**Example 6** *Consider the following MV-F-IEs* [8, 13]

$$\Lambda(\varrho, \sigma) = g(\varrho, \sigma) - \int\limits_0^\varrho \int\limits_0^1 (2z - 1)e^y \Lambda(y, z) dy dz, \tag{6.8}$$

*where* $g(\varrho, \sigma) = \sin(\varrho) + \sigma - \frac{1}{6}e^\varrho + \frac{1}{6}$. *The exact solution is* $\Lambda(\varrho, \sigma) = \sin(\varrho) + \sigma$.

In this example, in Table 7 we list a comparison between the AEs for Example 6 acquired in [13] and the results acquired by the introduced technique at various choices of N, M. Furthermore, in the Table 8, we introduce the MAEs for the problem presented in Example 6 for different values of N, M.

**Example 7** *Finally, we discuss the next non-linear MV-F-IEs* [14]

$$\Lambda(\varrho, \sigma) = g(\varrho, \sigma) - \int\limits_0^1 \int\limits_0^1 (\varrho\sigma + yz^2)\Lambda(y, z) dy dz - \int\limits_0^\sigma \int\limits_0^\varrho (\varrho + \sigma + y + z)(\Lambda(y, z))^2 dy dz \tag{6.9}$$

**Table 7. AEs for Example 6.**

| | MQs L-G-L [13] | | | |
|---|---|---|---|---|
| $(\varrho_i, s_i)$ | N = M = 2 | N = M = 3 | N = M = 4 | N = M = 5 |
| (0,0) | $3.44 \times 10^{-3}$ | $1.12 \times 10^{-3}$ | $5.25 \times 10^{-5}$ | $1.93 \times 10^{-5}$ |
| (0.1,0.1) | $3.54 \times 10^{-2}$ | $3.53 \times 10^{-2}$ | $3.50 \times 10^{-2}$ | $3.51 \times 10^{-2}$ |
| (0.2,0.2) | $7.20 \times 10^{-2}$ | $7.42 \times 10^{-2}$ | $7.38 \times 10^{-2}$ | $7.38 \times 10^{-2}$ |
| (0.3,0.3) | $1.14 \times 10^{-01}$ | $1.17 \times 10^{-01}$ | $1.17 \times 10^{-01}$ | $1.17 \times 10^{-01}$ |
| (0.4,0.4) | $1.62 \times 10^{-01}$ | $1.64 \times 10^{-01}$ | $1.64 \times 10^{-01}$ | $1.64 \times 10^{-01}$ |
| (0.5,0.5) | $2.16 \times 10^{-01}$ | $2.16 \times 10^{-01}$ | $2.16 \times 10^{-01}$ | $2.16 \times 10^{-01}$ |
| (0.6,0.6) | $2.76 \times 10^{-01}$ | $2.74 \times 10^{-01}$ | $2.74 \times 10^{-01}$ | $2.74 \times 10^{-01}$ |
| (0.7,0.7) | $3.41 \times 10^{-01}$ | $3.38 \times 10^{-01}$ | $3.38 \times 10^{-01}$ | $3.38 \times 10^{-01}$ |
| (0.8,0.8) | $4.11 \times 10^{-01}$ | $4.09 \times 10^{-01}$ | $4.09 \times 10^{-01}$ | $4.09 \times 10^{-01}$ |
| (0.9,0.9) | $4.86 \times 10^{-01}$ | $4.87 \times 10^{-01}$ | $4.87 \times 10^{-01}$ | $4.87 \times 10^{-01}$ |
| | Present Method | | | |
| $(\varrho_i, s_i)$ | N = M = 2 | N = M = 4 | N = M = 8 | N = M = 10 |
| (0,0) | $4.78 \times 10^{-3}$ | $1.47 \times 10^{-5}$ | $2.79 \times 10^{-7}$ | $1.07 \times 10^{-14}$ |
| (0.1,0.1) | $1.59 \times 10^{-3}$ | $1.45 \times 10^{-5}$ | $2.19 \times 10^{-7}$ | $7.41 \times 10^{-15}$ |
| (0.2,0.2) | $4.25 \times 10^{-3}$ | $9.85 \times 10^{-7}$ | $2.19 \times 10^{-7}$ | $7.51 \times 10^{-15}$ |
| (0.3,0.3) | $4.19 \times 10^{-3}$ | $1.29 \times 10^{-5}$ | $2.34 \times 10^{-7}$ | $1.03 \times 10^{-14}$ |
| (0.4,0.4) | $2.39 \times 10^{-3}$ | $1.23 \times 10^{-5}$ | $1.18 \times 10^{-7}$ | $8.38 \times 10^{-15}$ |
| (0.5,0.5) | $2.25 \times 10^{-4}$ | $1.84 \times 10^{-7}$ | $3.30 \times 10^{-7}$ | $4.16 \times 10^{-17}$ |
| (0.6,0.6) | $2.75 \times 10^{-3}$ | $1.17 \times 10^{-5}$ | $1.24 \times 10^{-7}$ | $8.34 \times 10^{-15}$ |
| (0.7,0.7) | $4.34 \times 10^{-3}$ | $1.21 \times 10^{-5}$ | $2.57 \times 10^{-7}$ | $1.02 \times 10^{-14}$ |
| (0.8,0.8) | $4.19 \times 10^{-3}$ | $1.16 \times 10^{-6}$ | $2.57 \times 10^{-7}$ | $7.13 \times 10^{-15}$ |
| (0.9,0.9) | $1.58 \times 10^{-3}$ | $1.36 \times 10^{-5}$ | $2.70 \times 10^{-7}$ | $7.13 \times 10^{-15}$ |

**Table 8. MAEs for Example 6.**

| $N = M$ | 2 | 4 | 8 | 10 |
|---------|---|---|---|-----|
| $MAE$ | $4.779 \times 10^{-3}$ | $1.473 \times 10^{-6}$ | $1.882 \times 10^{-11}$ | $1.069 \times 10^{-14}$ |

**Table 9. AEs for Example 7.**

| $(\varrho, \sigma)$ | Operational matrices [14] | | | Present Method | |
|---------------------|-----------|-----------|-----------|----------------|-----------|
|                     | $N = M = 4$ | $N = M = 8$ | $N = M = 16$ | $N = M = 2$ | $N = M = 4$ |
| (0,0) | $2.1639 \times 10^{-4}$ | $4.7124 \times 10^{-6}$ | $1.0025 \times 10^{-5}$ | $1.7137 \times 10^{-16}$ | $6.6987 \times 10^{-17}$ |
| (0.1,0.1) | $7.6440 \times 10^{-3}$ | $4.0972 \times 10^{-4}$ | $1.2786 \times 10^{-4}$ | $7.8063 \times 10^{-17}$ | $4.5103 \times 10^{-17}$ |
| (0.2,0.2) | $5.1937 \times 10^{-3}$ | $1.6343 \times 10^{-4}$ | $5.6840 \times 10^{-4}$ | $9.7145 \times 10^{-17}$ | $1.2490 \times 10^{-16}$ |
| (0.3,0.3) | $6.3855 \times 10^{-3}$ | $8.8137 \times 10^{-4}$ | $4.0802 \times 10^{-4}$ | $2.6368 \times 10^{-16}$ | $1.1102 \times 10^{-16}$ |
| (0.4,0.4) | $7.3341 \times 10^{-3}$ | $7.6984 \times 10^{-4}$ | $2.9045 \times 10^{-4}$ | $1.9429 \times 10^{-16}$ | $8.3267 \times 10^{-17}$ |
| (0.5,0.5) | $7.2081 \times 10^{-3}$ | $1.3335 \times 10^{-3}$ | $1.2740 \times 10^{-4}$ | $1.6653 \times 10^{-16}$ | $1.3878 \times 10^{-16}$ |
| (0.6,0.6) | $8.3994 \times 10^{-3}$ | $2.7182 \times 10^{-3}$ | $4.5862 \times 10^{-4}$ | $2.7756 \times 10^{-16}$ | $2.7756 \times 10^{-16}$ |
| (0.7,0.7) | $3.1165 \times 10^{-2}$ | $6.2178 \times 10^{-3}$ | $1.3792 \times 10^{-3}$ | $6.1062 \times 10^{-16}$ | $1.6653 \times 10^{-16}$ |
| (0.8,0.8) | $5.1521 \times 10^{-2}$ | $1.2029 \times 10^{-2}$ | $2.9430 \times 10^{-3}$ | $1.2212 \times 10^{-15}$ | $1.1102 \times 10^{-16}$ |
| (0.9,0.9) | $6.4482 \times 10^{-2}$ | $1.8815 \times 10^{-2}$ | $4.1130 \times 10^{-3}$ | $2.4415 \times 10^{-15}$ | $3.3307 \times 10^{-16}$ |

*The exact solution is $\Lambda(\varrho, \sigma) = \varrho^2 + 2\varrho\sigma$, where $g(\varrho, \sigma)$ is a given function extracted from the exact solution.*

To emphasize the high accuracy of our method for the two-dimensional problem, we offer a comparison between the AEs obtained in [14] and the results acquired by the proposed technique with several choices of $N$ and $M$ of Example 7 in Table 9. We observe that, a good approximation of the exact solution of the two-dimensional space MV-F-IEs is achieved for relatively small numbers of the collocation nodes $N$ and $M$.

## 7 Conclusion

In this paper, we have introduced an accurate and efficient numerical technique via the SJ-GL-C method for the spectral solutions for one- and two-dimensional MV-F-IEs. The exponential convergence of the spectral algorithm was investigated. According to the numerical results acquired in the preceding section, we have achieved a high-accuracy technique for solving multi-dimensional MV-F-IEs. In addition, the numerical results completely coincide with the theoretical results of the convergence analysis. The accuracy of the present scheme is compared with those of other results in the literature, which reveals the high accuracy and powerful of the current scheme. The algorithm is efficient, applicable to various operators, and extends to multi-dimensions, allowing for future research. In the future, we will discuss the stochastic integral equations as well as stochastic integro-differential equations.

## Acknowledgments

Cras egestas velit mauris, eu mollis turpis pellentesque sit amet. Interdum et malesuada fames ac ante ipsum primis in faucibus. Nam id pretium nisi. Sed ac quam id nisi malesuada congue. Sed interdum aliquet augue, at pellentesque quam rhoncus vitae.

## Author Contributions

**Data curation:** A. Z. Amin.

**Formal analysis:** A. Z. Amin.

**Investigation:** A. Z. Amin.

**Methodology:** A. Z. Amin, M. A. Abdelkawy, I. Hashim.

**Resources:** A. Z. Amin, M. A. Abdelkawy, I. Hashim.

**Software:** A. Z. Amin, M. A. Abdelkawy, I. Hashim.

**Validation:** A. Z. Amin.

**Visualization:** A. Z. Amin.

**Writing – original draft:** A. Z. Amin, A. K. Amin, M. A. Abdelkawy, A. A. Alluhaybi, I. Hashim.

**Writing – review & editing:** A. Z. Amin, A. K. Amin, M. A. Abdelkawy, A. A. Alluhaybi, I. Hashim.

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
