## [Decision Letter · Decision Letter 0]

8 Feb 2023

PONE-D-23-02279Spectral technique with convergence analysis for solving multi-dimensional mixed Volterra-Fredholm integral equationPLOS ONE

Dear Dr. Abdelkawy,

Thank you for submitting your manuscript to PLOS ONE. After careful consideration, we feel that it has merit but does not fully meet PLOS ONE’s publication criteria as it currently stands. Therefore, we invite you to submit a revised version of the manuscript that addresses the points raised during the review process.

We look forward to receiving your revised manuscript.

Kind regards,

Mahmoud A. Zaky

Academic Editor

PLOS ONE

Journal Requirements:

Additional Editor Comments:

1- The language should be polished.

2- In the title "multi-dimensional" but there are only one- and two dimensions problems. What about the high dimensions?

3- The novelty of the paper should be discussed in comparison with proposed high dimensional spectral methods for the same problems in literature.

4-Please add references for lemmas in section 5.1.

5- Please follow "?" in the manuscript and add missed references.

6 The list of references should be updated and its style should be unified.

Reviewers' comments:

Reviewer's Responses to Questions

**Comments to the Author**

1. Is the manuscript technically sound, and do the data support the conclusions?

Reviewer #1: Partly

Reviewer #2: Yes

2. Has the statistical analysis been performed appropriately and rigorously? 

Reviewer #1: N/A

Reviewer #2: Yes

3. Have the authors made all data underlying the findings in their manuscript fully available?

Reviewer #1: Yes

Reviewer #2: Yes

4. Is the manuscript presented in an intelligible fashion and written in standard English?

Reviewer #1: No

Reviewer #2: Yes

5. Review Comments to the Author

Reviewer #1: Herein, the authors offered a numerical scheme based on shifted Jacobi-Gauss collocation method for handling the mixed Volterra-Fredholm integral equations. The method transform the equation into a system of algebraic equations. The present algorithm is extended to solve the two-dimensional case. Convergence analysis for the present method is discussed. Various numerical examples are approached to demonstrate the applicability of the technique. I have read the text thoroughly, it needs a careful revision before the possibility of its potential publication in PONE, the following comments should be meticulously addressed, a point-by-point rebuttal letter is a must:

1. It is not good to boost citations by citing Refs via bulk criteria, see [1, 2, 3, 4, 5, 6, 7, 8], [24, 24, 25, 26, 27, 28], [29, 30, 31, 32, 33, 34, 35, 36, 37]. Please cite only relevant work; also use the cite package to present Refs as [1-4], not [1,2,3,4].

2. The main motivation of the method should be polished within the introductory section.

3. All properties in Section 2 are not new, it is not fair to do not cite a seminal text for these properties, The text of Shen et al. 2011, https://link.springer.com/book/10.1007/978-3-540-71041-7 is welcome.

4. For the two models in 3.1 and 3.2, please cite a Ref in which you picked these models from, also mention the regularity conditions on all functions parameters that ascertain the existence and uniqueness of the solutions, see for instance 10.22075/IJNAA.2015.179

5. The convergence rate of the unknown expansion coefficients was previously studied in depth in https://ijnao.um.ac.ir/article_25278.html;
https://doi.org/10.1016/j.camwa.2019.03.011;
https://doi.org/10.1007/s40314-018-0633-3. Please mention this at the beginning of Section 5.

6. Some Equations in Page 7 do not fit with the margins of the paper, please adopt.

7. Compare your work with https://doi.org/10.1007/s40065-019-0243-y.

8. Mention the platform used to write the codes as well as the specifications of the machine used to debug these codes.

9. Comment on future extensions in this direction in the conclusion section.

10. Proofread the whole manuscript to get rid of any typos or grammatical errors.

sss

Reviewer #2: The paper discusses the analysis and the numerical solution of mixed Volterra-Fredholm integral equations (MV-F-IEs). The numerical approach is based on shifted Jacobi-Gauss collocation (SJ-G-C) method. The proposed technique with shifted Jacobi-Gauss (SJ-G) nodes is applied to diminish the MV-F-IEs to a system of algebraic equations that are easily solved. The present algorithm is extended to solve the two-dimensional MV-F-IEs. Convergence analysis for the present method is discussed and confirms the exponential convergence of the spectral algorithm. Various numerical examples are approached to demonstrate the power and accuracy of the technique.

The work is meaningful and has sufficient novelty, although the same technique seems to be applied to other models by the co-authors. However, the analysis is technical and concrete, and the results are fascinating. The overall structure seems to be logical; definitions are given most of the time. Thus, I suggest the acceptance of the paper for publication. Before that, the authors should consider the following minor comments.

For the minor comments, see the attached file.

6. PLOS authors have the option to publish the peer review history of their article (what does this mean?). If published, this will include your full peer review and any attached files.

Reviewer #1: **Yes: **Youssri Hassan Youssri

Reviewer #2: **Yes: **A. K. Omran

<quillbot-extension-portal></quillbot-extension-portal>

---

## [Author Response · Author response to Decision Letter 0]

8 Mar 2023

Dear respected Editor

We verified certain that all tables were cited in the text, as seen below.

Table 6.1 mentioned in Lines 212 and 218.

Table 6.2 mentioned in Line 229.

Table 6.3 mentioned in Line234.

Table 6.4 mentioned in Line 240.

Responses to referees

I) With respect to the fi\frst referee (Reviewer: 1):

1. We carefully checked our manuscript and corrected the errors through-

out the paper.

2. The main motivation of the method added in the introductory sec-

tion.

3. We checked the references list and inserted additional references.

4. We have reviewed and corrected all of the reviewer's required changes.

5. All additions were highlighted in red color.

II) With respect to the second referee (Reviewer: 2):

The main advantages of the present study are:

1. We carefully checked our manuscript and corrected the errors through-

out the paper.

2. The main motivation of the method added in the introductory sec-

tion.

3. We inserted references for the properties mentioned in Section 2.

4. For the two models in 3.1 and 3.2, We inserted suitable references.

5. We compared our work with https://doi.org/10.1007/s40065-019-0243-

y.(ex3).

6. We mentioned the platform used to write the codes as well as the

speci\fcations of the machine used to debug these codes.

7. We cited only relevant work; also used the cite package.

8. we inserted a Comment on future extensions in this direction in the

conclusion section.

9. All additions were highlighted in red color.

II) With respect to the Editor:

1. We carefully checked our manuscript and corrected the errors through-

out the paper.

2. The main motivation of the method added in the introductory sec-

tion.

3. We replaced "multi-dimensional" by "one- and two-dimensional"in

the title.

4. We inserted a comparison with proposed high dimensional spectral

methods for the same problems in literature.

5. We added references for lemmas in section 5.1.

6. We checked the references list and inserted additional references.

7. The reference list has been updated, and the style has been uni\fed.

8. All additions were highlighted in red color.

---

## [Decision Letter · Decision Letter 1]

16 Mar 2023

Spectral technique with convergence analysis for solving one and two-dimensional mixed Volterra-Fredholm integral equation

PONE-D-23-02279R1

Dear Dr. Abdelkawy,

We’re pleased to inform you that your manuscript has been judged scientifically suitable for publication and will be formally accepted for publication once it meets all outstanding technical requirements.

Kind regards,

Mahmoud A. Zaky

Academic Editor

PLOS ONE

<quillbot-extension-portal></quillbot-extension-portal>

---

## [Editor Report · Acceptance letter]

28 Mar 2023

PONE-D-23-02279R1 

Spectral technique with convergence analysis for solving one and two-dimensional mixed Volterra-Fredholm integral equation 

Dear Dr. Abdelkawy:

I'm pleased to inform you that your manuscript has been deemed suitable for publication in PLOS ONE. Congratulations! Your manuscript is now with our production department. 

Kind regards, 

on behalf of

Dr. Mahmoud A. Zaky 

Academic Editor

PLOS ONE